# Outpatient Department Visits and Mortality with Various Causes Attributable to Ambient Air Pollution in the Eastern Economic Corridor of Thailand

**DOI:** 10.3390/ijerph19137683

**Published:** 2022-06-23

**Authors:** Khanut Thongphunchung, Panita Charoensuk, Sutida U-tapan, Wassana Loonsamrong, Arthit Phosri, Wiriya Mahikul

**Affiliations:** 1Health Impact Assessment Division, Department of Health, Ministry of Public Health, Nonthaburi 11000, Thailand; khanut.th@gmail.com (K.T.); panita.charoensuk@gmail.com (P.C.); usutida@gmail.com (S.U.-t.); loonsamrong.blue@gmail.com (W.L.); 2Department of Environmental Health Sciences, Faculty of Public Health, Mahidol University, Bangkok 10400, Thailand; arthit.pho@mahidol.ac.th; 3Center of Excellence on Environmental Health and Toxicology, Bangkok 10400, Thailand; 4Princess Srisavangavadhana College of Medicine, Chulabhorn Royal Academy, Bangkok 10210, Thailand

**Keywords:** air pollution, particulate matter, outpatient department visits, mortality, Eastern Economic Corridor, case-crossover study, conditional Poisson model

## Abstract

The Eastern Economic Corridor in Thailand is undergoing development, but industrial activities are causing serious air pollution. This study aimed to examine the effects of particulate matter (PM_10_), fine particulate matter (PM_2.5_), SO_2_, NO_2_, O_3_, and CO on outpatient department (OPD) visits and mortality with various causes in the Eastern Economic Corridor, Thailand between 2013 and 2019 using a case-crossover design and conditional Poisson model. The corresponding burden of disease due to air pollution exposure was calculated. A 1 µg/m^3^ increase in the PM_10_ was associated with significant increases in OPD visits for circulatory diseases (0.22, 95% CI 0.01, 0.34), respiratory diseases (0.21, 95% CI 0.13, 0.28), and skin and subcutaneous tissue diseases (0.18, 95% CI 0.10, 0.26). By contrast, a 1 µg/m^3^ increase in the PM_10_ was associated with significant increases in mortality from skin and subcutaneous tissue diseases (0.79, 95% CI 0.04, 1.56). A 1 µg/m^3^ increase in PM_2.5_ was associated with significant increases in mortality from circulatory diseases (0.75, 95% CI 0.20, 1.34), respiratory diseases (0.82, 95% CI 0.02, 1.63), and skin and subcutaneous tissue diseases (2.91, 95% CI 0.99, 4.86). The highest OPD burden was for circulatory diseases. Respiratory effects were attributed to PM_10_ exceeding the national ambient air quality standards (NAAQS) of Thailand (120 μg/m^3^). The highest morbidity burden was for skin and subcutaneous tissue diseases attributed to PM_2.5_ concentrations that exceeded the NAAQs (50 μg/m^3^). PM pollution in the EEC could strongly contribute to OPD visits and morbidity from various diseases. Preventing PM_10_ concentrations from being higher than 60 µg/m^3^ could decrease OPD visits by more than 33,265 and 29,813 for circulatory and respiratory diseases, respectively. Our study suggests that such pollution increases the risks of OPD visits and mortality in various causes in the Thai EEC. Reducing the ambient air pollution concentration of NAAQSs in Thailand could reduce the health effect on the Thai population.

## 1. Introduction

The development of the Eastern Economic Corridor (EEC) in Thailand is a 20-year strategy involving improving infrastructure to increase investment and improve transportation and human development. The aim is to allow Thailand to compete with wealthier countries with more knowledge-based economies and to become a high-income country by 2036. An area-based development initiative has been established with the aim of revitalizing the eastern seaboard of Thailand over a 30-year period to encourage innovative and high-value industries. Development was initially focused on the three eastern provinces, Chachoengsao, Chonburi, and Rayong [1]. Various industrial processes can cause pollutants to be emitted (including into the air) and result in waste products that pose risks to humans and the environment. More serious air pollution tends to be caused by manufacturing industries than other industries because manufacturing industries are energy-intensive and produce large amounts of pollutants [2]. Diesel engines used for transportation can emit large amounts of particulate matter (PM) and carbon dioxide and other greenhouse gases that contribute to climate change.

Air pollution is an important global threat to the environment and human health. PM pollution in air causes 7 × 10^6^ premature deaths of humans around the world each year [3]. PM is a complex mixture of solid and liquid particles of organic and inorganic substances suspended in the air. The main components of PM are ammonia, black carbon, mineral dust, nitrates, sodium chloride, sulfates, and water [4]. Some of the main components of PM are classed as carcinogenic to humans by the International Agency for Research on Cancer [5]. PM with diameters ≤10 μm is most damaging to health, and PM with diameters ≤2.5 μm (PM_2.5_) is of most concern because it can penetrate deep into the lungs and affect both the respiratory and vascular systems. It has been found that an increase of 10 μg/m^3^ in the O_3_, NO_2_, SO_2_, or PM_10_ concentration or an increase of 1 mg/m^3^ in the CO concentration over a 1 d period in Bangkok, Thailand is associated with an increase in hospital admissions for cardiovascular problems [6]. Air pollution causes and exacerbates a wide range of diseases, including asthma, heart disease, lung cancer, respiratory disease, and stroke [7,8]. The effects of PM on human health are becoming increasingly concerning because epidemiological studies are indicating more links between exposure to PM and adverse health effects. Both short-term and long-term exposure to PM is associated with adverse health effects, including acute outcomes (mortality) [9,10], hospital admissions [6], visits to emergency and outpatient departments (OPDs) [11], absences from school or work [12], acute symptoms [13], chronic disease outcomes (chronic cardiovascular diseases, chronic cardio-respiratory diseases, and lung cancer) [14], and reproductive effects (fetal deaths, low birth weights, pre-term deliveries) [14,15,16]. It is, therefore, important to improve our understanding of the relationships between adverse health effects and air pollution in the EEC of Thailand.

The Map Ta Phut industrial estate in Rayong is an important source of pollution in the EEC. The Map Ta Phut industrial estate contains petrochemical and heavy industry plants and is a source of heavy metals and organophosphates to the environment [17]. Improving our understanding of the relationships between adverse health effects and exposure to air pollution will support the development of policies and strategies for decreasing such adverse health effects and protecting human health. This study was performed to improve our understanding of the relationships between adverse health effects and air pollution in the EEC. The main objective of this study aimed to examine the association between exposure to daily changes in ambient air pollution and OPD visits, and mortality for a spectrum of causes in the overall population in the EEC. 

## 2. Materials and Methods

### 2.1. Study Area

Thailand Eastern Economic Corridor (EEC) is an area-based development initiative, initially focused on the 3 Eastern provinces of Thailand because of their strategic locations near deep seaports and natural resources in the Gulf of Thailand, including Rayong, Chonburi, and Chachoengsao provinces, and spans approximately 13,266 sq.km., as shown in Figure 1. The EEC plays an important role in Thailand’s fast-growing economy in relation to the aim of transforming the country into a hub for technological manufacturing and services with strong connectivity to its association of Southeast Asian Nations (ASEAN) neighbors by land, rail, sea, and air transportation, where it has designated 28 promoted zones and spans approximately 138.84 sq.km. to be a location for the 12 targeted industries, including (1) next-generation automotive, (2) intelligent electronics, (3) automation & robotics, (4) digital, (5) advanced agriculture & biotechnology, (6) food for the future, (7) medical & comprehensive healthcare, (8) high-value & medical tourism, (9) aviation & logistics, (10) biofuel & biochemical, (11) defense, and (12) education & human resource development. Moreover, the EEC has established promoted zones for specific industries and new cities into 7 areas with more than 19.25 sq.km. to serve prospective investors around the world [18]. There were 10,745 factories registered in the EEC at the end of 2019. The most common types of factories in this area were metal, automotive and auto part, plastic, machinery and metal work, and chemical plants [19]. All residents of the EEC area at the end of 2019 were considered to be the target population. The total population of the EEC area at the end of 2019 was 3.01 million people [20]. Most of the population lives in urban areas where there is a monitoring station. Therefore, the population density was high in urban areas. 

### 2.2. Air Pollution and Meteorological Data

Ambient air pollutant concentration and meteorological measurements were made at fixed ambient air monitoring stations operated by the Thailand Pollution Control Department. The concentrations of ambient air pollutants (PM_10_, PM_2.5_, CO, NO_2_, SO_2_, and O_3_) and meteorological data (temperature and relative humidity) for the period 1 October 2013 to 31 December 2019 (2283 d) were obtained from the Pollution Control Department. The pollutant data were used to represent exposure of the general population of the study area to the pollutants. The locations of the air quality monitoring stations are shown in Figure 1. Nine stations in the EEC were operating during the study period. One station was in Chachoengsao, three were in Chonburi, and five were in Rayong. The daily mean concentration of each air pollutant at each monitoring station was calculated from data collected in each 24-h period during the study period. If air pollutant concentration data for <18 h (75% of a 24-h period) were available for a day, the data for that day were defined as missing. If there were multiple monitoring stations in the same province, the daily mean concentration of each air pollutant was calculated using data from all of the monitoring stations and used to indicate human exposure to the air pollutant in that province. If air pollution and meteorological data were available for >75% of a 24-h period, the missing data were replaced with values estimated using the expectation-maximization (EM) algorithm, which uses the method under the assumption of multivariate normal distribution. This algorithm is specially tailored for climate data with missing measurements from several monitors along a given region [21]. If air pollution and meteorological data were available for <75% of a 24-h period, the missing data were excluded from the study.

### 2.3. Outpatient Department Visits and Mortality Data

There were 1226 hospitals, including government hospitals, private hospitals, primary care health units, and private clinics, located in the ECC, 239 in Chachoengsao, 430 in Chonburi, and 557 in Rayong. Data of patient who has the Universal Coverage Scheme (UCS) were reported to the National Health Security Office (NHSO). The data for OPD visits for various causes (including cardiovascular diseases, respiratory diseases, and skin and subcutaneous tissue diseases) in the EEC during the period air pollution and meteorological data were acquired by NHSO. The visit date, the date of birth, age, and sex of the patient, and the primary diagnosis code from the International Classification of Diseases 10th Revision were acquired. Mortality data derived from information reported on death certificates during the same period were supplied by the Strategy and Planning Division of the Thailand Ministry of Public Health. The date of birth, age, and sex of each deceased person and the International Classification of Diseases 10th Revision cause-of-death code were used. 

### 2.4. Study Design and Statistical Analyses

A case-crossover experimental design was used to investigate associations between short-term exposure to ambient air pollutants and the risks of OPD visits and mortality from various causes. Each patient or death was used as a case and their own control, and the concentrations of air pollutants in the hazard period were compared with the concentrations of air pollutants in the control period, the same day of the week in different months or years [22,23]. Factors that would have acted as confounders of the health outcomes from the health characteristics of the subjects were removed [24], including the effects of other covariates, such as seasonal effects [25,26]. The effects of potential confounding factors that varied within a week, such as weekday–weekend differences in air pollution and visits to OPDs, were decreased by the experimental design. Control days before and after the case day were selected to remove potential bias caused by long-term air pollution trends [27]. Matching by month controlled for potential seasonal confounding factors and matching by year controlled for long-term trends [24,28]. Each case, therefore, had three or four controls. For example, if the index day was a Sunday, another Sunday in the same month was used as a control. A separate dataset was created for each disease group and each province. 

A conditional Poisson model was used to assess associations between short-term exposure to each ambient air pollutant and the number of OPD visits and mortality from various causes. The model was adjusted for potential non-linear confounding effects of temperature and relative humidity that could affect symptoms and air pollutant concentrations by including natural cubic splines with three degrees of freedom [10,29,30]. The model equation was
Log(E(Y_t_)) = α + β_1_X_t_ + ns(temp_t_, 3df) + ns(RH_t_, 3df) + province_t_ + stratum_t_,(1)
where log(E(Y_t_)) is the predicted number of OPD visits or deaths on day t, Y_t_ is the actual number of OPD visits or deaths on day t, α is the y-intercept, β_1_ is the regression coefficient per unit increase in the air pollutant concentration, and the values of the vector of coefficient were presented in Appendix A, β1 was calculated by the maximum likelihood method, X_t_ is the air pollutant concentration in province i on day t, ns() is the natural cubic spline with a defined number of degrees of freedom (df), and temp_t_ and RH_t_ are the temperature and relative humidity on day t, province_t_ is indicator variable on day t, stratum is a matched case–control within the same day of the week in the same calendar month of the same year on day t, respectively.

The estimates were made as relative risks and 95% confidence intervals (95% CIs), regression coefficients β_1_, and standard errors for the coefficients for OPD visits or deaths from various causes associated with unit increases in the daily concentrations of the air pollutants. The relative risks were calculated separately for each pollutant.

The number of OPD visits and mortality burden caused by exposure to air pollutants were estimated using national ambient air quality standards (NAAQS) for Thailand as reference concentrations. Comparisons were made using 50% and 25% of the NAAQS as reference concentrations. The attributable fraction (AF) and attributable number (AN) were calculated using the effect estimates determined as described above using the equations
AF_x, t_ = 1 − exp(−β_x, t_)(2)
and
AN_x, t_ = AF_x, t_ X n_i,_(3)
where AF_x, t_ is the fraction of OPD visits or deaths attributable to causes associated with air pollutant x on day t, AN_x, t_ is the number of OPD visits or deaths attributable to air pollutant x on day t, β_x, t_ is the coefficient of the association multiplied by the difference between the observed air pollutant x concentration and the reference concentrations of air pollutant x on day t, and X n_i_ is the number of OPD visits or deaths.

The numbers of OPD visits or deaths with a cause specific to exposure to each air pollutant were aggregates of the ANs for each day and the AF was used as an estimate of the proportion of OPD visits or deaths from causes attributed to the air pollutant (calculated by dividing the AN by the total number of OPD visits or deaths from each cause). The 95% CIs were calculated by performing Monte Carlo simulations assuming that the data followed a normal distribution. All analyses were performed using R software.

## 3. Results

Summary statistics of the air pollutant and meteorological variable data for each province in the EEC are shown in Table 1. The daily mean NO_2_ concentration, PM_10_ concentration, CO concentration, and temperature during the study period were higher in Chonburi (12.68 ± 5.45 ppb, 40.29 ± 18.92 µg/m^3^, 0.81 ± 0.34 ppm, and 28.60 ± 1.85 °C, respectively) than the other provinces. The daily mean SO_2_ concentration, PM_2.5_ concentration, and relative humidity were higher in Rayong (3.44 ± 1.66 ppb, 21.56 ± 13.84 µg/m^3^, and 76.22% ± 8.03%, respectively) than the other provinces. The daily mean O_3_ concentration was higher in Chachoengsao (42.15 ± 15.46 ppb) than the other provinces. The NO_2_ concentration, PM_10_ concentration, O_3_ concentration, temperature, and relative humidity varied in all three provinces. 

Summary statistics for OPD visits and mortality from various causes for each province in the EEC are shown in Table 2 and Table 3. There were 12,698,730 OPD visits for the causes of interest during the study period. There were the most OPD visits in Chonburi (8,026,570), next most in Rayong (4,265,950), and fewest in Chachoengsao (406,210). The number of OPD visits in all three provinces decreased in the order circulatory diseases (6,160,104), respiratory diseases (5,323,737), and skin and subcutaneous tissue diseases (1,214,889). There were 31,406 deaths from the causes of interest. There were the most deaths in Chonburi (15,376), next most in Rayong (12,851), and fewest in Chachoengsao (3179). The number of deaths in all three provinces decreased in the order of circulatory diseases (22,872), respiratory diseases (8141), and skin and subcutaneous tissue diseases (393). 

The associations between unit increases in the air pollutant concentrations and OPD visits for specific causes are shown in Table 4. The percent changes and 95% CI for unit increases in the air pollutant concentrations are shown. A 1 µg/m^3^ increase in the PM_10_ concentration was associated with significant increases in OPD visits for circulatory diseases (0.22, 95% CI 0.01, 0.34), respiratory diseases (0.21, 95% CI 0.13, 0.28), and skin and subcutaneous tissue diseases (0.18, 95% CI 0.10, 0.26). A 1 µg/m^3^ increase in the PM_2.5_ concentration was not significantly associated with increases in OPD visits for the diseases of interest. A 1 ppb increase in the NO_2_ concentration was associated with significant increases in OPD visits for circulatory diseases (1.38, 95% CI 1.04, 1.72), respiratory diseases (0.93, 95% CI 0.73, 1.14), and skin and subcutaneous tissue diseases (1.03, 95% CI 0.79, 1.27). A 1 ppb increase in the SO_2_ concentration was associated with significant increases in OPD visits for respiratory diseases (1.23, 95% CI 0.73, 1.74) and skin and subcutaneous tissue diseases (0.71, 95% CI 0.15, 1.27). A 1 ppb increase in the O_3_ concentration was associated with increases in OPD visits for acute upper respiratory infections (0.08, 95% CI 0.01, 0.15) but not with increases in OPD visits for other diseases. No significant associations were found between the CO concentration and OPD visits for the diseases of interest.

The associations between unit increases in air pollutant concentrations and mortality are shown in Table 5. There was an association between a 1 µg/m^3^ increase in the PM_10_ concentration and increases in mortality from skin and subcutaneous tissue diseases (0.79, 95% CI 0.04, 1.56), but there were no associations between the PM_10_ concentration and mortality from respiratory diseases (0.16, 95% CI −0.23, 0.55) and circulatory diseases (0.22, 95% CI −0.07, 0.52). A 1 µg/m^3^ increase in the PM_2.5_ concentration was associated with significant increases in mortality from circulatory diseases (0.75, 95% CI 0.20, 1.34), respiratory diseases (0.82, 95% CI 0.02, 1.63), and skin and subcutaneous tissue diseases (2.91, 95% CI 0.99, 4.86). A 1 ppb increase in the NO_2_ concentration was associated with significant increases in mortality from circulatory diseases (1.17, 95% CI 0.26, 2.07) and skin and subcutaneous tissue diseases (5.22, 95% CI 2.81, 7.68) but not with mortality from respiratory diseases (0.93, 95% CI −0.28, 2.16). No significant associations were found between mortality and a 1 ppb increase in the SO_2_ concentration. Positive associations were found between mortality from circulatory diseases and a 1 ppb increase in the O_3_ concentration (0.27, 95% CI 0.00, 0.55) and a 1 ppm increase in the CO concentration (30.96, 95% CI 8.88, 57.53).

Adverse health effects, such as OPD visits for circulatory diseases, can occur when PM_10_ concentrations are higher than the NAAQS for Thailand of 120 μg/m^3^ and mortality caused by circulatory diseases can occur when PM_2.5_ concentrations are higher than 50 μg/m^3^ [31]. The AFs for OPD visits for all diseases in the EEC attributed to PM_10_ are shown in Table 6, Table 7, Table 8, Table 9, Table 10, Table 11, Table 12, Table 13, Table 14 and Table 15. The PM_2.5_ AF for OPD visits for circulatory diseases was 0.01% (95% CI 0.00%, 0.02%), and the PM_2.5_ AF for OPD visits for respiratory problems was 0.01% (95% CI 0.00%, 0.02%). The PM_2.5_ AF for mortality from circulatory diseases was 0.20% (95% CI 0.01%, 0.45%), and the PM_2.5_ AF for mortality from respiratory problems was 0.20% (95% CIs 0.01%, 0.45%). However, PM_2.5_ affected skin and subcutaneous tissue diseases little. The PM_10_ AN for OPD visits for circulatory diseases was 616 (95% CI 62, 1232), and the PM_2.5_ AN for mortality from circulatory diseases was 46 (95% CI 2, 103). The estimated AFs and ANs for OPD visits and mortality for various causes in the EEC and the three provinces in the EEC calculated using the NAAQSs, 50% of the NAAQSs, and 25% of the NAAQSs as references are shown in Table 6, Table 7, Table 8, Table 9, Table 10, Table 11, Table 12, Table 13, Table 14 and Table 15. A hypothetical intervention that capped the PM_10_ concentration at a maximum of 60 μg/m^3^ would have prevented 33,265 OPD visits (95% CI 1232, 62,833) for circulatory diseases and 29,813 OPD visits (95% CI 1065, 67,612) for respiratory diseases.

## 4. Discussion

Air pollution poses risks to human health around the world. In previous epidemiological studies, it has been found that air pollution can be related to respiratory diseases and cardiovascular diseases [29,32,33,34,35,36]. Most previous relevant studies in Thailand were performed at a small number of locations and, therefore, provided limited evidence for associations between air pollution and health effects [6,9,10,11,37,38,39,40,41]. We found higher daily mean NO_2_ concentrations, PM_10_ concentrations, CO concentrations, and temperatures during the study period in Chonburi than the other provinces. Similar results were found in a previous study [38]. Chonburi is a rapidly growing economy and rising in population and urbanization compared with other provinces. Considering a stationary source as a factory and industrial estate, there were 5418 factories registered and 11 industrial estates in the province, and the highest proportions of the factory and industrial estate were 50.42% and 52.38%, respectively [18,19]. Moreover, in terms of mobile sources as a vehicle, there were 1,628,404 cars registered (57.76%) [42]. These point sources could lead to higher air pollution concentration than other provinces. OPD visits and deaths in all three provinces in the EEC were caused by, in decreasing order, circulatory diseases, respiratory diseases, and skin and subcutaneous tissue diseases. Similar results were found in a previous study in Bangkok, Thailand [6]. We assessed OPD visits and mortality for various causes that were attributable to ambient air pollution in the EEC from 2013 to 2019. Daily PM_10_ concentrations were significantly associated with increased OPD visits for circulatory diseases, respiratory diseases, and skin and subcutaneous tissue diseases, and daily PM_2.5_ concentrations were significantly associated with death from circulatory diseases, respiratory diseases, and skin and subcutaneous tissue diseases. In a previous study in Thailand, the highest AF of disease burden for all causes of mortality in adults attributable to PM_2.5_ was 10%, which was higher than the fraction attributable to any other air pollutant [43]. Increased risks of specific respiratory diseases were also associated with exposure to PM. OPD visits for upper respiratory tract diseases were associated with exposure to PM_10_, and mortality from chronic lower respiratory diseases was associated with exposure to PM_2.5_. Similar results were found in previous epidemiological studies performed in various countries; at lag0, the RR of respiratory outpatients increased by 0.37%, with a 10 μg/m^3^ increase in PM_2.5_ in Shanghai, China [44]. In addition, the studies performed in Thailand showed the association between a 10 μg/m^3^ increase in PM_10_ concentration and an increase in cardiovascular mortality (1.90 95% CI 1.00, 2.80) [10]. In Bangkok Metropolitan Region, the odds ratios of respiratory OPD among children per 10 µg/m^3^ increase in PM_10_ and PM_2.5_ were 1.03 (95% CI 1.02, 1.03) and 1.03 (95% CI 1.01, 1.06), respectively [11]. The study of the impact of PM on daily mortality in Bangkok indicated a statistically significant association between PM_10_ and cardiovascular and respiratory mortality; a 10 µg/m^3^ change in daily PM_10_ is associated with a 1–2% and a 3–6% increase, respectively [39]. The estimated effect seems higher than this study because the estimation was explored in 10 µg/m^3^.

The higher relative risk of short-term exposure to air pollution on mortality may be explained by the susceptible population reasons. Older adults and children or people with heart (or lung) disease are subject to much stronger risk from particles than other people [45]. Several epidemiologic studies have shown the evidence of short-term exposure to air pollution on mortality. The estimated effects of short-term exposure to PM_10_ and PM_2.5_ from meta-analysis studies of 17 Chinese cities showing a 10 μg/m^3^ increase in PM_2.5_ concentration associated with increases in mortality due to circulatory diseases and respiratory diseases were 0.63% (95% CI 0.35%, 0.91%) and 0.75% (95% CI 0.39%, 1.11%), respectively [46]. In a study in the Netherlands, a 10 μg/m^3^ increase in PM_2.5_ was related with an increase of 1.60% (95% CI 0.40%, 2.90%) in respiratory mortality and 1.10% (95% CI 0.20%, 1.90%) in cardiovascular at lag0 [47]. Short-term PM_2.5_ exposures were significantly associated with increased all-cause mortality among residents of New England ≥ 65 years of age; 2.14% (95% CI 1.38%, 2.89%) increases were estimated for each 10 μg/m^3^ increase in short-term exposure (2 day), even for particle exposure levels below the standards established by the United States and the World Health Organization (WHO) (25 μg/m^3^ of daily average PM_2.5_) [48]. The study in Thailand showed a high cardiovascular mortality percent change of a 10 μg/m^3^ increase in PM_10_ [1.90% (95% CI 1.00%, 2.80%)] at lag0 [10]. The mechanisms of PM on triggering respiratory diseases showed that the oxidative stress and inflammation process drive adverse effects related to respiratory diseases. Particularly, PM could trigger an inflammatory response, potentially leading to airway obstruction, affecting the gas-exchange process, and exacerbating pre-existing conditions of respiratory disease [39,49]. 

The observed effects of exposure to PM on OPD visits and mortality from skin and subcutaneous tissue diseases could be explained toxicologically. The skin is part of the primary immune system and is always exposed to air. Many industrial chemicals can be absorbed through the skin and cause local toxicity. Particles may cause oxidative stress by causing reactive oxygen species to be produced, and this can cause certain skin diseases [50].

We found that PM contributed to OPD visits, as was found in previous studies. It was found in a study performed in Dongguan, China that 8.32% of total respiratory morbidity and 9.77% of asthma morbidity could be attributed to exposure to PM_2.5_ [51]. It was found in a study performed in Chengdu, China that 3.52% of total hospital admissions for respiratory diseases, 2.51% of hospital admissions for circulatory diseases, and 4.28% of hospital admissions for skin and subcutaneous tissue diseases could be attributed to high PM_10_ concentrations and that 3.84% of hospital admissions for circulatory diseases could be attributed to high PM_10_ concentrations using the World Health Organization air quality guidelines (24-h mean PM_10_ and PM_2.5_ concentrations of 50 and 25 µg/m^3^, respectively) [52].

Although a 1 ppb increase in the NO_2_ concentration was associated with significant increases in OPD visits for circulatory diseases, respiratory diseases, and skin and subcutaneous tissue diseases and a 1 ppb increase in the SO_2_ concentration was associated with significant increases in OPD visits for respiratory diseases and skin and subcutaneous tissue diseases, attributable fractions and attributable numbers of OPD visits for circulatory diseases, respiratory diseases, and skin and subcutaneous tissue diseases attributable to NO_2_ and SO_2_ were small in amount in this study. The associations between these pollutions and health outcomes in the EEC should be studied in the future with different approaches.

The estimated associations between OPD visits and morbidity attributable to ambient air pollutants will be important to public health authorities because they will support health policy revisions and the development of preventative interventions. The AFs and ANs indicate the proportions of disease burdens in the population caused by exposure to excessive ambient PM concentrations that could be prevented if exposure to PM was prevented. The AFs and ANs could, therefore, be used to set air quality standards. The AFs and ANs provide more information than relative risks and excess risks because the proportions and numbers of mortalities and morbidities caused by exposure to ambient PM are quantified [53]. AFs and ANs are, therefore, suitable for estimating the potential health benefits of establishing air pollution standards and guidelines [54]. In this study, we found that preventing PM_10_ concentrations from being higher than 60 µg/m^3^ could decrease OPD visits for circulatory and respiratory diseases. The PM_10_ concentration of 60 µg/m^3^ could, therefore, be used as an air quality standard for Thailand.

The study had some limitations. First, PM_2.5_ data for Chachoengsao and Chonburi were insufficiently available and inappropriate for the study period. Therefore, those data were excluded from the study. Unfortunately, data on PM_2.5_ were available only in Rayong. Consequently, the association between PM_2.5_ and OPD for a spectrum of causes was not statistically significant. More PM_2.5_ data will be required for further study. Second, the available ambient air pollution and meteorological variable from only one fixed-site air monitoring station in Chachoengsao was assigned as an exposure level to the population. These exposure misclassifications can cause bias in the measurement of effect toward the null value, producing an underestimate of the effect; however, it is unavoidable [55,56]. Third, the daily mean concentrations of the air pollutants from all the monitoring stations in a province were used to indicate human exposure to the air pollutants in that province. This may have given incorrect estimates for human exposure to the air pollutants. Fourth, personal exposure could be affected by the duration and intensity of outdoor activities and other lifestyle factors. Fifth, the results represented all populations in the EEC but not a specific subgroup in age or gender. The risky group or the susceptible group should be analysed in the future. Sixth, particles are complex mixtures with various chemical and physical characteristics and toxicities. Further research will be required to identify the most dangerous components of PM and the most sensitive population subgroups. Geostatistical analysis should be performed in a future study to estimate exposure at locations not near air quality monitoring stations.

## 5. Conclusions

The effects of air pollutants on various causes of OPD visits and mortality in the EEC were assessed. Significant associations were found between the concentrations of ambient air pollutants (PM_10_, PM_2.5_, NO_2_, SO_2_, O_3_, and CO) and adverse health effects. These pollutants can cause hospital OPD visits because of respiratory diseases and cause marked morbidity related to circulatory and respiratory diseases in the EEC. The results provided new epidemiological evidence to support policy revisions and the development of preventative interventions to minimize the adverse health effects caused by air pollutants and protect human health.

## Figures and Tables

**Figure 1 ijerph-19-07683-f001:**
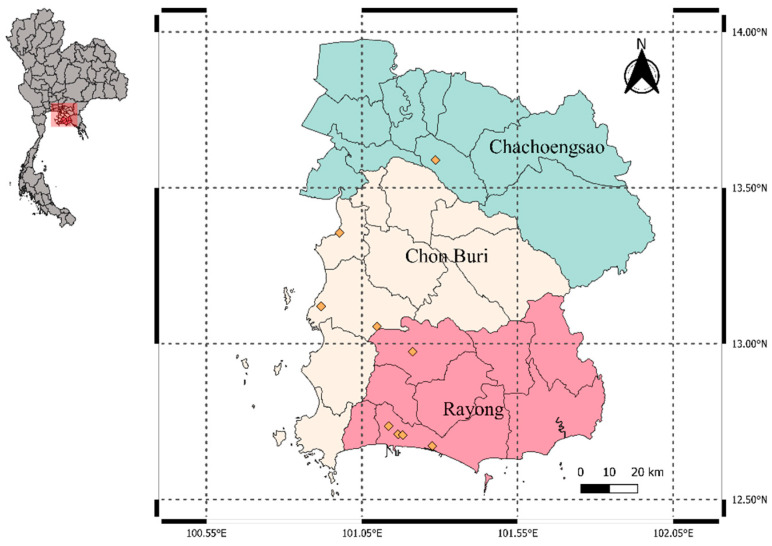
Study area with orange diamonds indicating air quality monitoring stations.

**Table 1 ijerph-19-07683-t001:** Summary statistics for the air pollutant concentrations and meteorological variables for the three provinces in the Eastern Economic Corridor of Thailand between 2013 and 2019.

Variable		Mean ± SD	
	Chachoengsao	Chonburi	Rayong
Air pollutant			
PM_10_ (µg/m^3^)	36.89 ± 18.04	40.29 ± 18.92	39.97 ± 18.12
PM_2.5_ (µg/m^3^)	NA	NA	21.56 ± 13.84
NO_2_ (ppb)	8.35 ± 4.06	12.68 ± 5.45	11.39 ± 4.55
SO_2_ (ppb)	2.19 ± 2.09	2.08 ± 1.67	3.44 ± 1.66
O_3_ (ppb)	42.15 ± 15.46	39.91 ± 16.83	34.21 ± 15.72
CO (ppm)	NA	0.81 ± 0.34	0.72 ± 0.25
Meteorological variable			
Temperature (°C)	28.06 ± 1.77	28.60 ± 1.85	28.42 ± 1.75
Relative humidity (%)	NA	72.74 ± 10.17	76.22 ± 8.03

SD = standard deviation. NA = data not available.

**Table 2 ijerph-19-07683-t002:** Summary statistics for outpatient department (OPD) visits with various causes in the three provinces in the Eastern Economic Corridor of Thailand between 2013 and 2019.

Cause		Number of OPD Visits			
ICD-10	Diagnostic Category	Chachoengsao	Chonburi	Rayong	Total
I00-I99	Circulatory diseases	389,225	4,152,470	2,007,634	6,160,104
I26-I28	Pulmonary heart disease and diseases of pulmonary circulation	11,475	194,173	72,426	278,074
J00-J99	Respiratory diseases	341,399	3,165,359	1,816,979	5,323,737
J00-J06	Acute upper respiratory infections	277,225	2,322,587	1,388,759	3,988,571
J09-J18	Influenza and pneumonia	5231	114,988	50,066	170,285
J20-J22	Acute lower respiratory infections	21,785	177,860	92,200	291,845
J30-J39	Diseases of upper respiratory tract	4655	177,557	90,561	272,773
J40-J47	Chronic lower respiratory diseases	31,611	349,943	184,496	566,050
L00-L99	Skin and subcutaneous tissue diseases	64,811	708,741	441,337	1,214,889
L20-L30	Dermatitis and eczema	36,740	291,994	181,651	510,385
	Total	406,210	8,026,570	4,265,950	12,698,730

Note: NA = no data available for the study period. ICD-10 = International Classification of Diseases 10th Revision.

**Table 3 ijerph-19-07683-t003:** Summary statistics for the number of deaths with various causes in the three provinces in the Eastern Economic Corridor of Thailand between 2013 and 2019.

Cause		Number of Deaths			
ICD-10	Diagnostic Category	Chachoengsao	Chonburi	Rayong	Total
I00-I99	Circulatory diseases	2055	11,684	9133	22,872
I26-I28	Pulmonary heart disease and diseases of pulmonary circulation	129	1493	1317	2939
J00-J99	Respiratory diseases	1109	3459	3573	8141
J00-J06	Acute upper respiratory infections	20	23	15	58
J09-J18	Influenza and pneumonia	150	1655	1388	3193
J20-J22	Acute lower respiratory infections	8	8	202	218
J30-J39	Diseases of upper respiratory tract	NA	11	9	20
J40-J47	Chronic lower respiratory diseases	64	394	376	834
L00-L99	Skin and subcutaneous tissue diseases	15	233	145	393
L20-L30	Dermatitis and eczema	NA	15	1	16
	Total	3179	15,376	12,851	31,406

Note: NA = no data available for the study period. ICD-10 = International Classification of Diseases 10th Revision.

**Table 4 ijerph-19-07683-t004:** Associations between unit increase in pollutant concentrations and outpatient department visits for specific causes.

Diagnostic Category	RR (95% CI) per Unit Increase in Concentration					
	PM_10_ (µg/m^3^)	PM_2.5_ (µg/m^3^)	NO_2_ (ppb)	SO_2_ (ppb)	O_3_ (ppb)	CO (ppm)
Circulatory diseases	1.0022 * (1.0010, 1.0034)	0.9999 (0.9977, 1.0020)	1.0138 * (1.0104, 1.0172)	1.0060 (0.9979, 1.0143)	0.9998 (0.9987, 1.0009)	1.0336 (0.9641, 1.1081)
Pulmonary heart disease and diseases of pulmonary circulation	1.0025 * (1.0011, 1.0040)	1.0009 (0.9983, 1.0034)	1.0137 * (1.0097, 1.0178)	1.0071 (0.9972, 1.0172)	1.0001 (0.9987, 1.0014)	1.0572 (0.9737, 1.1478)
Respiratory diseases	1.0021 * (1.0013, 1.0028)	1.0003 (0.9990, 1.0016)	1.0093 * (1.0073, 1.0114)	1.0123 * (1.0073, 1.0174)	1.0006 (0.9999, 1.0013)	1.0350 (0.9918, 1.0801)
Acute upper respiratory infections	1.0022 * (1.0014, 1.0029)	1.0002 (0.9990, 1.0015)	1.0092 * (1.0071, 1.0113)	1.0145 * (1.0095, 1.0195)	1.0008 * (1.0001, 1.0015)	1.0364 (0.9932, 1.0815)
Influenza and pneumonia	1.0019 * (1.0007, 1.0030)	1.0012 (0.9988, 1.0035)	1.0092 * (1.0057, 1.0126)	1.0021 (0.9937, 1.0105)	1.0005 (0.9995, 1.0016)	1.0508 (0.9803, 1.1264)
Acute lower respiratory infections	1.0016 * (1.0007, 1.0025)	1.0008 (0.9990, 1.0027)	1.0072 * (1.0045, 1.0098)	1.0103 * (1.0040, 1.0166)	0.9997 (0.9989, 1.0006)	1.0219 (0.9690, 1.0777)
Diseases of upper respiratory tract	1.0024 * (1.0012, 1.0036)	1.0005 (0.9981, 1.0029)	1.0128 * (1.0093, 1.0164)	1.0122 * (1.0035, 1.0210)	0.9999 (0.9988, 1.0011)	1.0538 (0.9792, 1.1341)
Chronic lower respiratory diseases	1.0015 * (1.0005, 1.0025)	1.0005 (0.9985, 1.0026)	1.0105 * (1.0076, 1.0134)	1.0019 (0.9950, 1.0089)	1.0000 (0.9990, 1.0009)	1.0165 (0.9574, 1.0792)
Skin and subcutaneous tissue diseases	1.0018 * (1.0010, 1.0026)	1.0005 (0.9991, 1.0018)	1.0103 * (1.0079, 1.0127)	1.0071 * (1.0015, 1.0127)	1.0000 (0.9992, 1.0008)	1.0433 (0.9933, 1.0958)
Dermatitis and eczema	1.0022 * (1.0013, 1.0032)	1.0007 (0.9992, 1.0023)	1.0121 * (1.0093, 1.0148)	1.0076 * (1.0013, 1.0140)	1.0001 (0.9993, 1.0010)	1.0520 (0.9952, 1.1121)

Notes: RR = relative risk. * = significant at *p* < 0.05.

**Table 5 ijerph-19-07683-t005:** Associations between unit increase in pollutant concentrations and mortality from specific causes.

Diagnostic Category	RR (95% CI) per Unit Increase in Concentration					
	PM_10_ (µg/m^3^)	PM_2.5_ (µg/m^3^)	NO_2_ (ppb)	SO_2_ (ppb)	O_3_ (ppb)	CO (ppm)
Circulatory diseases	1.0022 (0.9993, 1.0052)	1.0075 * (1.0020, 1.0130)	1.0117 * (1.0026, 1.0207)	1.0155 (0.9941, 1.0373)	1.0027 * (1.0000, 1.0055)	1.3096 * (1.0888, 1.5753)
Pulmonary heart disease and diseases of pulmonary circulation	1.0035 (0.9982, 1.0087)	1.0156 * (1.0040, 1.0274)	1.0112 (0.9951, 1.0275)	1.0253 (0.9866, 1.0656)	0.9978 (0.9930, 1.0027)	1.3299 (0.9385, 1.8845)
Respiratory diseases	1.0016 (0.9977, 1.0055)	1.0082 * (1.0002, 1.0163)	1.0093 (0.9972, 1.0216)	0.9924 (0.9645, 1.0211)	1.0006 (0.9970, 1.0042)	1.0670 (0.8248, 1.3802)
Acute upper respiratory infections	1.0335 * (1.0220, 1.0451)	1.0605 * (1.0236, 1.0988)	1.1093 * (1.0515, 1.1702)	1.2333 * (1.1233, 1.3541)	0.9933 (0.9845, 1.0022)	NA
Influenza and pneumonia	1.0024 (0.9976, 1.0072)	1.0133 * (1.0031, 1.0235)	1.0154 * (1.0006, 1.0305)	0.9748 (0.9394, 1.0116)	0.9999 (0.9955, 1.0044)	1.2224 (0.8881, 1.6827)
Acute lower respiratory infections	1.0956 * (1.0760, 1.1156)	1.0472 * (1.0179, 1.0773)	1.0574 (0.9970, 1.1214)	1.5144 * (1.3456, 1.7044)	1.0088 (0.9952, 1.0227)	NA
Diseases of upper respiratory tract	1.0736 * (1.0519, 1.0957)	NA	0.9941 (0.9553, 1.0346)	1.0726 (0.9992, 1.1513)	0.9905 (0.9749, 1.0063)	NA
Chronic lower respiratory diseases	0.9987 (0.9918, 1.0057)	1.0198 * (1.0025, 1.0374)	1.0045 (0.9830, 1.0265)	0.9818 (0.9290, 1.0377)	1.0019 (0.9957, 1.0081)	0.7888 (0.4895, 1.2713)
Skin and subcutaneous tissue diseases	1.0079 * (1.0004, 1.0156)	1.0291 * (1.0099, 1.0486)	1.0522 * (1.0281, 1.0768)	1.0281 (0.9710, 1.0886)	1.0014 (0.9947, 1.0081)	0.9389 (0.4932, 1.7876)
Dermatitis and eczema	1.0257 * (1.0045, 1.0473)	0.9871 (0.8416, 1.1576)	1.2086 * (1.0864, 1.3445)	NA	NA	NA

Note: NA = no data available. RR = relative risk. * = significant at *p* < 0.05.

**Table 6 ijerph-19-07683-t006:** Attributable fractions and attributable numbers of outpatient department (OPD) visits and mortalities for circulatory diseases caused by air pollutant concentrations in the Eastern Economic Corridor of Thailand exceeded the national ambient air quality standard (NAAQS) (100%), 50% of the NAAQS (50%), and 25% of the NAAQS (25%) between 2013 and 2019.

Pollutant (NAAQS)	Attributable Fraction (95% CIs)			Attributable Number (95% CIs)		
	100%	50%	25%	100%	50%	25%
PM_10_ (120 µg/m^3^)						
OPD visits	0.01 (0.00, 0.02)	0.54 (0.02, 1.02)	2.83 (0.09, 6.27)	616 (62, 1232)	33,265 (1232, 62,833)	174,331 (5544, 386,238)
Mortality	-	-	-	-	-	-
PM_2.5_ (50 µg/m^3^)						
OPD visits	-	-	-	-	-	-
Mortality	0.20 (0.01, 0.45)	1.41 (0.04, 3.15)	3.14 (0.10, 6.99)	46 (2, 103)	322 (9, 720)	718 (23, 1599)
NO_2_ (170 ppb)						
OPD visits	0.00 (0.00, 0.00)	0.00 (0.00, 0.00)	0.01 (0.00, 0.02)	154 (6, 246)	154 (6, 246)	616 (62, 1232)
Mortality	0.00 (0.00, 0.00)	0.00 (0.00, 0.00)	0.00 (0.00, 0.00)	1 (1, 1)	1 (1, 1)	1 (1, 1)
SO_2_ (120 ppb)						
OPD visits	-	-	-	-	-	-
Mortality	-	-	-	-	-	-
O_3_ (70 ppb)						
OPD visits	-	-	-	-	-	-
Mortality	0.16 (0.01, 0.36)	2.21 (0.07, 4.84)	5.46 (0.17, 12.19)	37 (2, 82)	505 (16, 1107)	1249 (39, 2788)
CO (9 ppm)						
OPD visits	-	-	-	-	-	-
Mortality	0.00 (0.00, 0.00)	0.00 (0.00, 0.00)	0.01 (0.00, 0.02)	1 (1, 1)	1 (1, 1)	2 (1, 5)

Note: - means no significant association.

**Table 7 ijerph-19-07683-t007:** Attributable fractions and attributable numbers of outpatient department (OPD) visits and mortalities for pulmonary heart disease and diseases of pulmonary circulation caused by air pollutant concentrations in the Eastern Economic Corridor of Thailand exceeded the national ambient air quality standard (NAAQS) (100%), 50% of the NAAQS (50%), and 25% of the NAAQS (25%) between 2013 and 2019.

Pollutant (NAAQS)	Attributable Fraction (95% CIs)			Attributable Number (95% CIs)		
	100%	50%	25%	100%	50%	25%
PM_10_ (120 µg/m^3^)						
OPD visits	0.01 (0.00, 0.02)	0.63 (0.02, 1.41)	3.29 (0.10, 7.50)	28 (3, 56)	1752 (56, 3921)	9149 (278, 20,856)
Mortality	-	-	-	-	-	-
PM_2.5_ (50 µg/m^3^)						
OPD visits	-	-	-	-	-	-
Mortality	0.31 (0.01, 0.69)	2.59 (0.08, 5.77)	6.16 (0.19, 13.83)	9 (1, 20)	76 (2, 170)	181 (6, 406)
NO_2_ (170 ppb)						
OPD visits	0.00 (0.00, 0.00)	0.00 (0.00, 0.00)	0.01 (0.00, 0.02)	5 (1, 7)	7 (1, 11)	28 (1, 56)
Mortality	-	-	-	-	-	-
SO_2_ (120 ppb)						
OPD visits	-	-	-	-	-	-
Mortality	-	-	-	-	-	-
O_3_ (70 ppb)						
OPD visits	-	-	-	-	-	-
Mortality	-	-	-	-	-	-
CO (9 ppm)						
OPD visits	-	-	-	-	-	-
Mortality	-	-	-	-	-	-

Note: - means no significant association.

**Table 8 ijerph-19-07683-t008:** Attributable fractions and attributable numbers of outpatient department (OPD) visits and mortalities for respiratory diseases caused by air pollutant concentrations in the Eastern Economic Corridor of Thailand exceeded the national ambient air quality standard (NAAQS) (100%), 50% of the NAAQS (50%), and 25% of the NAAQS (25%) between 2013 and 2019.

Pollutant (NAAQS)	Attributable Fraction (95% CIs)			Attributable Number (95% CIs)		
	100%	50%	25%	100%	50%	25%
PM_10_ (120 µg/m^3^)						
OPD visits	0.01 (0.00, 0.02)	0.56 (0.02, 1.27)	2.69 (0.08, 6.06)	532 (53, 1065)	29,813 (1065, 67,612)	143,209 (4259, 322,618)
Mortality	-	-	-	-	-	-
PM_2.5_ (50 µg/m^3^)						
OPD visits	-	-	-	-	-	-
Mortality	0.20 (0.01, 0.45)	1.47 (0.05, 3.26)	3.41 (0.11, 7.66)	16 (1, 37)	120 (4, 265)	278 (9, 624)
NO_2_ (170 ppb)						
OPD visits	0.00 (0.00, 0.00)	0.00 (0.00, 0.00)	0.00 (0.00, 0.00)	15 (1, 27)	46 (11, 80)	76 (27, 160)
Mortality	-	-	-	-	-	-
SO_2_ (120 ppb)						
OPD visits	0.00 (0.00, 0.00)	0.00 (0.00, 0.00)	0.00 (0.00, 0.00)	27 (5, 53)	80 (27, 106)	160 (53, 213)
Mortality	-	-	-	-	-	-
O_3_ (70 ppb)						
OPD visits	-	-	-	-	-	-
Mortality	-	-	-	-	-	-
CO (9 ppm)						
OPD visits	-	-	-	-	-	-
Mortality	-	-	-	-	-	-

Note: - means no significant association.

**Table 9 ijerph-19-07683-t009:** Attributable fractions and attributable numbers of outpatient department (OPD) visits and mortalities for acute upper respiratory infections caused by air pollutant concentrations in the Eastern Economic Corridor of Thailand exceeded the national ambient air quality standard (NAAQS) (100%), 50% of the NAAQS (50%), and 25% of the NAAQS (25%) between 2013 and 2019.

Pollutant (NAAQS)	Attributable Fraction (95% CIs)			Attributable Number (95% CIs)		
	100%	50%	25%	100%	50%	25%
PM_10_ (120 µg/m^3^)						
OPD visits	0.01 (0.00–0.02)	0.60 (0.02, 1.33)	2.82 (0.09, 6.37)	399 (177, 798)	23,931 (798, 53,048)	112,478 (3590, 254,072)
Mortality	0.00 (0.00, 0.00)	12.07 (0.39, 27.10)	29.31 (0.89, 66.18)	1 (1, 1)	7 (1, 16)	17 (1, 38)
PM_2.5_ (50 µg/m^3^)						
OPD visits	-	-	-	-	-	-
Mortality	0.00 (0.00, 0.00)	5.17 (0.18, 11.43)	8.62 (0.27, 19.70)	1 (1, 1)	3 (1, 7)	5 (1, 11)
NO_2_ (170 ppb)						
OPD visits	0.00 (0.00, 0.00)	0.00 (0.00, 0.00)	0.00 (0.00, 0.00)	4 (1, 7)	30 (21, 56)	59 (29, 128)
Mortality	0.00 (0.00, 0.00)	0.00 (0.00, 0.00)	0.00 (0.00, 0.00)	1 (1, 1)	1 (1, 1)	1 (1, 1)
SO_2_ (120 ppb)						
OPD visits	0.00 (0.00, 0.00)	0.00 (0.00, 0.00)	0.00 (0.00, 0.00)	9 (4, 16)	53 (25, 78)	75 (38, 154)
Mortality	0.00 (0.00, 0.00)	0.00 (0.00, 0.00)	0.00 (0.00, 0.00)	1 (1, 1)	1 (1, 1)	1 (1, 1)
O_3_ (70 ppb)						
OPD visits	0.05 (0.00, 0.11)	0.68 (0.02, 1.54)	1.68 (0.06, 3.73)	1994 (177, 4387)	27,122 (798, 61,424)	67,008 (2393, 148,774)
Mortality	-	-	-	-	-	-
CO (9 ppm)						
OPD visits	-	-	-	-	-	-
Mortality	-	-	-	-	-	-

Note: - means no significant association.

**Table 10 ijerph-19-07683-t010:** Attributable fractions and attributable numbers of outpatient department (OPD) visits and mortalities for influenza and pneumonia caused by air pollutant concentrations in the Eastern Economic Corridor of Thailand exceeded the national ambient air quality standard (NAAQS) (100%), 50% of the NAAQS (50%), and 25% of the NAAQS (25%) between 2013 and 2019.

Pollutant (NAAQS)	Attributable Fraction (95% CIs)			Attributable Number (95% CIs)		
	100%	50%	25%	100%	50%	25%
PM_10_ (120 µg/m^3^)						
OPD visits	0.00 (0.00, 0.00)	0.39 (0.01, 0.87)	2.27 (0.07, 5.11)	3 (1, 7)	664 (17, 1481)	3865 (119, 8702)
Mortality	-	-	-	-	-	-
PM_2.5_ (50 µg/m^3^)						
OPD visits	-	-	-	-	-	-
Mortality	0.41 (0.01, 0.92)	2.47 (0.07, 5.55)	5.57 (0.17, 12.01)	13 (1, 29)	79 (2, 177)	178 (5, 383)
NO_2_ (170 ppb)						
OPD visits	0.00 (0.00, 0.00)	0.00 (0.00, 0.00)	0.00 (0.00, 0.00)	1 (1, 1)	2 (1, 4)	3 (1, 7)
Mortality	0.00 (0.00, 0.00)	0.00 (0.00, 0.00)	0.00 (0.00, 0.00)	1 (1, 1)	1 (1, 1)	1 (1, 1)
SO_2_ (120 ppb)						
OPD visits	-	-	-	-	-	-
Mortality	-	-	-	-	-	-
O_3_ (70 ppb)						
OPD visits	-	-	-	-	-	-
Mortality	-	-	-	-	-	-
CO (9 ppm)						
OPD visits	-	-	-	-	-	-
Mortality	-	-	-	-	-	-

Note: - means no significant association.

**Table 11 ijerph-19-07683-t011:** Attributable fractions and attributable numbers of outpatient department (OPD) visits and mortalities for acute lower respiratory infections caused by air pollutant concentrations in the Eastern Economic Corridor of Thailand exceeded the national ambient air quality standard (NAAQS) (100%), 50% of the NAAQS (50%), and 25% of the NAAQS (25%) between 2013 and 2019.

Pollutant (NAAQS)	Attributable Fraction (95% CIs)			Attributable Number (95% CIs)		
	100%	50%	25%	100%	50%	25%
PM_10_ (120 µg/m^3^)						
OPD visits	0.01 (0.00, 0.02)	0.42 (0.01, 0.94)	2.06 (0.07, 4.62)	29 (1, 58)	1226 (58, 2743)	6012 (20, 13,483)
Mortality	0.00 (0.00, 0.00)	4.59 (0.15, 10.41)	16.06 (0.58, 36.35)	1 (1, 1)	10 (1, 23)	35 (1, 79)
PM_2.5_ (50 µg/m^3^)						
OPD visits	-	-	-	-	-	-
Mortality	0.46 (0.02, 1.04)	2.29 (0.08, 5.08)	13.30 (0.39, 30.06)	1 (1, 2)	5 (1, 11)	29 (1, 66)
NO_2_ (170 ppb)						
OPD	0.00 (0.00, 0.00)	0.00 (0.00, 0.00)	0.00 (0.00, 0.00)	1 (1, 1)	1 (1, 1)	3 (1, 9)
Mortality	-	-	-	-	-	-
SO_2_ (120 ppb)						
OPD visits	0.00 (0.00, 0.00)	0.00 (0.00, 0.00)	0.00 (0.00, 0.00)	1 (1, 1)	1 (1, 1)	3 (1, 12)
Mortality	0.00 (0.00, 0.00)	0.00 (0.00, 0.00)	0.00 (0.00, 0.00)	1 (1, 1)	1 (1, 1)	1 (1, 1)
O_3_ (70 ppb)						
OPD visits	-	-	-	-	-	-
Mortality	-	-	-	-	-	-
CO (9 ppm)						
OPD visits	-	-	-	-	-	-
Mortality	-	-	-	-	-	-

Note: - means no significant association.

**Table 12 ijerph-19-07683-t012:** Attributable fractions and attributable numbers of outpatient department (OPD) visits and mortalities for diseases of the upper respiratory tract caused by air pollutant concentrations in the Eastern Economic Corridor of Thailand exceeded the national ambient air quality standard (NAAQS) (100%), 50% of the NAAQS (50%), and 25% of the NAAQS (25%) between 2013 and 2019.

Pollutant (NAAQS)	Attributable Fraction (95% CIs)			Attributable Number (95% CIs)		
	100%	50%	25%	100%	50%	25%
PM_10_ (120 µg/m^3^)						
OPD visits	0.01 (0.00–0.02)	0.63 (0.02, 1.43)	3.15 (0.09, 6.96)	27 (1, 55)	1718 (55, 3901)	8592 (245, 18,985)
Mortality	0.00 (0.00, 0.00)	10.00 (0.32, 21.78)	40.00 (1.37, 89.76)	1 (1, 1)	1 (1, 4)	1 (1, 8)
PM_2.5_ (50 µg/m^3^)						
OPD visits	-	-	-	-	-	-
Mortality	-	-	-	-	-	-
NO_2_ (170 ppb)						
OPD visits	0.00 (0.00, 0.00)	0.00 (0.00, 0.00)	0.01 (0.00, 0.02)	1 (1, 1)	1 (1, 7)	27 (1, 55)
Mortality	-	-	-	-	-	-
SO_2_ (120 ppb)						
OPD visits	0.00 (0.00, 0.00)	0.00 (0.00, 0.00)	0.00 (0.00, 0.00)	1 (1, 1)	1 (1, 9)	1 (1, 11)
Mortality	-	-	-	-	-	-
O_3_ (70 ppb)						
OPD visits	-	-	-	-	-	-
Mortality	-	-	-	-	-	-
CO (9 ppm)						
OPD visits	-	-	-	-	-	-
Mortality	-	-	-	-	-	-

Note: - means no significant association.

**Table 13 ijerph-19-07683-t013:** Attributable fractions and attributable numbers of outpatient department (OPD) visits and mortalities for chronic lower respiratory diseases caused by air pollutant concentrations in the Eastern Economic Corridor of Thailand exceeded the national ambient air quality standard (NAAQS) (100%), 50% of the NAAQS (50%), and 25% of the NAAQS (25%) between 2013 and 2019.

Pollutant (NAAQS)	Attributable Fraction (95% CIs)			Attributable Number (95% CIs)		
	100%	50%	25%	100%	50%	25%
PM_10_ (120 µg/m^3^)						
OPD visits	0.00 (0.00, 0.00)	0.38 (0.01, 0.86)	1.92 (0.06, 4.31)	10 (1, 23)	2151 (57, 4868)	10,868 (340, 24,397)
Mortality	-	-	-	-	-	-
PM_2.5_ (50 µg/m^3^)						
OPD visits	-	-	-	-	-	-
Mortality	0.36 (0.01, 0.80)	2.40 (0.08, 5.49)	6.24 (0.20, 14.33)	3 (1, 7)	20 (1, 46)	52 (2, 120)
NO_2_ (170 ppb)						
OPD visits	0.00 (0.00, 0.00)	0.00 (0.00, 0.00)	0.00 (0.00, 0.00)	1 (1, 1)	4 (1, 8)	8 (1, 20)
Mortality	-	-	-	-	-	-
SO_2_ (120 ppb)						
OPD visits	-	-	-	-	-	-
Mortality	-	-	-	-	-	-
O_3_ (70 ppb)						
OPD visits	-	-	-	-	-	-
Mortality	-	-	-	-	-	-
CO (9 ppm)						
OPD visits	-	-	-	-	-	-
Mortality	-	-	-	-	-	-

Note: - means no significant association.

**Table 14 ijerph-19-07683-t014:** Attributable fractions and attributable numbers of outpatient department (OPD) visits and mortalities for skin and subcutaneous tissue diseases caused by air pollutant concentrations in the Eastern Economic Corridor of Thailand exceeded the national ambient air quality standard (NAAQS) (100%), 50% of the NAAQS (50%), and 25% of the NAAQS (25%) between 2013 and 2019.

Pollutant (NAAQS)	Attributable Fraction (95% CIs)			Attributable Number (95% CIs)		
	100%	50%	25%	100%	50%	25%
PM_10_ (120 µg/m^3^)						
OPD visits	0.00 (0.00, 0.00)	0.38 (0.01, 0.86)	2.07 (0.07, 4.63)	24 (1, 49)	4617 (121, 10,448)	25,148 (850, 56,249)
Mortality	0.51 (0.02, 1.15)	2.54 (0.09, 30.12)	10.18 (0.32, 23.03)	2 (1, 5)	10 (1, 118)	40 (1, 91)
PM_2.5_ (50 µg/m^3^)						
OPD visits	-	-	-	-	-	-
Mortality	0.51 (0.02, 1.13)	2.80 (0.09, 6.30)	6.62 (0.20,14.70)	2 (1, 4)	11 (1, 25)	26 (1, 58)
NO_2_ (170 ppb)						
OPD visits	0.00 (0.00, 0.00)	0.00 (0.00, 0.00)	0.00 (0.00, 0.00)	1 (1, 1)	1 (1, 1)	24 (1, 49)
Mortality	0.00 (0.00, 0.00)	0.00 (0.00, 0.00)	0.25 (0.01, 0.56)	1 (1, 1)	1 (1, 1)	1 (1, 2)
SO_2_ (120 ppb)						
OPD visits	0.00 (0.00, 0.00)	0.00 (0.00, 0.00)	0.00 (0.00, 0.00)	1 (1, 1)	1 (1, 1)	24 (1, 49)
Mortality	-	-	-	-	-	-
O_3_ (70 ppb)						
OPD visits	-	-	-	-	-	-
Mortality	-	-	-	-	-	-
CO (9 ppm)						
OPD visits	-	-	-	-	-	-
Mortality	-	-	-	-	-	-

Note: - means no significant association.

**Table 15 ijerph-19-07683-t015:** Attributable fractions and attributable numbers of outpatient department (OPD) visits and mortalities for dermatitis and eczema caused by air pollutant concentrations in the Eastern Economic Corridor of Thailand exceeded the national ambient air quality standard (NAAQS) (100%), 50% of the NAAQS (50%), and 25% of the NAAQS (25%) between 2013 and 2019.

Pollutant (NAAQS)	Attributable Fraction (95% CIs)			Attributable Number (95% CIs)		
	100%	50%	25%	100%	50%	25%
PM_10_ (120 µg/m^3^)						
OPD visits	0.01 (0.00, 0.02)	0.47 (0.01, 1.06)	2.54 (0.08, 5.65)	51 (1, 102)	2399 (51, 5410)	12,964 (408, 28,837)
Mortality	0.00 (0.00, 0.00)	0.00 (0.00, 0.00)	18.75 (0.63, 42.98)	1 (1, 1)	1 (1, 1)	3 (1, 7)
PM_2.5_ (50 µg/m^3^)						
OPD visits	-	-	-	-	-	-
Mortality	-	-	-	-	-	-
NO_2_ (170 ppb)						
OPD visits	0.00 (0.00, 0.00)	0.00 (0.00, 0.00)	0.00 (0.00, 0.00)	1 (1, 1)	1 (1, 1)	13 (1, 23)
Mortality	0.00 (0.00, 0.00)	0.00 (0.00, 0.00)	0.00 (0.00, 0.00)	1 (1, 1)	1 (1, 1)	1 (1, 1)
SO_2_ (120 ppb)						
OPD visits	0.00 (0.00, 0.00)	0.00 (0.00, 0.00)	0.00 (0.00, 0.00)	1 (1, 1)	1 (1, 1)	13 (1, 23)
Mortality	-	-	-	-	-	-
O_3_ (70 ppb)						
OPD visits	-	-	-	-	-	-
Mortality	-	-	-	-	-	-
CO (9 ppm)						
OPD visits	-	-	-	-	-	-
Mortality	-	-	-	-	-	-

Note: - means no significant association.

## Data Availability

The data presented in this study are available on request from the corresponding author. The data are not publicly available due to ethical.

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
