# Peer review of "Outpatient Department Visits and Mortality with Various Causes Attributable to Ambient Air Pollution in the Eastern Economic Corridor of Thailand"

_ijerph, 2022, doi:10.3390/ijerph19137683_

Round 1
Reviewer 1 Report
Thongphunchung et al. assessed the association between air pollution and outpatient department visits/ mortality rate in the Eastern Economic Corridor in Thailand. Although these association has been found various times in other studies, ranging from different regions and countries, the manuscript Thongphunchung et al presented still provides valuable information about the newly developed economic hub in Thailand, and thus shed light on the potential adverse effect of these fast-growing national engines.
Comments:
Linen65, both short-term and long-term exposure to PM “is” associated…
Line 73, I believe the manuscript could benefit from a better description of what industries are in the Eastern Economic Corridor, at least a brief introduction/survey. One of the major differences of this manuscript compared to other similar studies is that the studying region is a newly developed, fast-growing economical special zone in southeast Asia.
Line 89, what is the size of the Eastern Economic Corridor region?
Line 90, there is only one site for air pollution monitoring in Chachoengsao. Is this sufficient/accurate for the analysis for that region? Does the population surveyed overlap with that specific site or is there a proportion of the population away from that area? What is the variation with the sites in the other two regions?
Line 108, how many/ percentage of days or data points are labeled as “missing” based on these criteria? Can the authors also elaborate on how the EM algorithm was used to impute the missing values and why would this be optimal? What statistic/distribution model was used as an assumption of pollution values?
Line 120, how many hospitals (for each region) were involved in this study?
Line 122, stated that the collected data contains age and gender information. However, in the subsequent analysis, no analysis was shown on related categories ( eg, pediatric visits vs adult visits, male vs female). Is there any specific reason such analysis was not performed?
Line 148, can the authors elaborate on why these variables (eg temperate, humidity) are sufficient to control the effects of air pollution on hospital visits? How about local populations etc.
Line 197, “the” fewest in Chachoengsao….
Line 202, in the order “of” circulatory diseases…
Line 245, table 4 title unit increase in pollutant concentrations… (no “the”), same for table 5 title.
Line 246, table 4. In this table, the authors presented the short-term association of air pollution to various hospital visits, with the relative risk values. Since there are some similar literature that also investigated this association in Thailand, how does the finding compare to other regions in Thailand or southeast Asia? Among those associations, is there anything specific in the Eastern Economic Corridor region? For example, Is the relative risk higher in the Eastern Economic Corridor region compared to other places?
Line249, table 5. Since the association evaluated was focused on the short-term exposure, using the case-crossover method, I’m surprised that the relative risk is still high and significant to mortality rates. This means exposure to increased air pollution as short as one day, could lead to direct death. Has the same short-term association of mortality been found in other literature? If not or rarely, does this means the current model used in the paper is possibly exaggerating the effect in some way?
Line 250, following the same question, if the authors analyze the data with another model, eg. Looking at lag days (polynomial distributed lag models etc), does the association (hospital visits and mortality) still hold?
Line 289, table 9 title, Thailand “exceeded” the national ambient air quality standard…
Author Response
Response to Reviewer 1 Comments
Comments and Suggestions for Authors
Thongphunchung et al. assessed the association between air pollution and outpatient department visits/ mortality rate in the Eastern Economic Corridor in Thailand. Although these association has been found various times in other studies, ranging from different regions and countries, the manuscript Thongphunchung et al presented still provides valuable information about the newly developed economic hub in Thailand, and thus shed light on the potential adverse effect of these fast-growing national engines.
Point 1:
Linen65, both short-term and long-term exposure to PM “is” associated…
Response 1:
Thank you for the insightful comments. We agreed with the reviewer and edited as suggestion as shown on the revised manuscript on line 109
Point 2:
Line 73, I believe the manuscript could benefit from a better description of what industries are in the Eastern Economic Corridor, at least a brief introduction/survey. One of the major differences of this manuscript compared to other similar studies is that the studying region is a newly developed, fast-growing economical special zone in southeast Asia.
Response 2:
We have further reviewed and revised the sentence as follows;
“Thailand Eastern Economic Corridor (EEC) is an area-based development initiative, initially focused on the 3 Eastern provinces of Thailand because of their strategic locations near deep seaports and natural resources in the Gulf of Thailand, including Rayong, Chonburi, and Chachoengsao provinces and spans approximately 13,266 sq.km., as shown in Figure 1. The EEC plays an important role in Thailand's fast-growing economy aim at transforming the country into a hub for technological manufacturing and services with strong connectivity to its association of Southeast Asian Nations (ASEAN) neighbors by land, rail, sea and air transportation, where has designated 28 promoted zones, spans approximately 138.84 sq.km., to be a location for the 12 targeted industries including 1) next-generation automotive 2) intelligent electronics 3) automation & robotics 4) digital 5) advanced agriculture & biotechnology 6) food for the future 7) medical & comprehensive healthcare 8) high-value & medical tourism 9) aviation & logistics 10) biofuel & biochemical 11) defense and 12) education & human resource development. Moreover, the EEC has established promoted zones for specific industries and new cities into 7 areas with more than 19.25 sq.km. to serve prospective investors around the world [18]. There were 8,141 factories registered in the EEC at the end of 2019. The most common types of factories in this area were metal, automotive and auto part, plastic, machinery and metal work, and chemical plants [19]. All residents of the EEC area at the end of 2019 were considered to be the target population. The total population of the EEC area at the end of 2019 was 3.01 million people. Most of the population lives in urban areas where have a monitoring station. Therefore, the population density was high in urban areas.”
as shown on the revised manuscript on line 129
Point 3:
Line 89, what is the size of the Eastern Economic Corridor region?
Response 3:
Thailand Eastern Economic Corridor (EEC) is an area-based development initiative, initially focused on the 3 Eastern provinces of Thailand because of their strategic locations near deep seaports and natural resources in the Gulf of Thailand, including Rayong, Chonburi, and Chachoengsao provinces and spans approximately 13,266 sq.km.
We added the sentences to explain the size of the Eastern Economic Corridor region as follows;
“Thailand Eastern Economic Corridor (EEC) is an area-based development initiative, initially focused on the 3 Eastern provinces of Thailand because of their strategic locations near deep seaports and natural resources in the Gulf of Thailand, including Rayong, Chonburi, and Chachoengsao provinces and spans approximately 13,266 sq.km. ...”
as shown on the revised manuscript on line 129
Point 4:
Line 90, there is only one site for air pollution monitoring in Chachoengsao. Is this sufficient/accurate for the analysis for that region? Does the population surveyed overlap with that specific site or is there a proportion of the population away from that area? What is the variation with the sites in the other two regions?
Response 4:
We agreed with the reviewer about the insufficient of one site for air pollution monitoring in Chachoengsao.
There was one of the limitations, the available ambient air pollution and meteorological variable from only one fixed-site air monitoring station in Chachoengsao was assigned as an exposure level to the population. These would be exposure misclassification causes measures of effect to be biased toward the null value. Misclassification produces an underestimate of the effect, whereas misclassification can result in bias either toward or away from the null value, is unavoidable.
- National Academies of Sciences, Engineering, and Medicine. 1997. Environmental Epidemiology, Volume 2: Use of the Gray Literature and Other Data in Environmental Epidemiology. Washington, DC: The National Academies Press. https://doi.org/10.17226/5804.
- Zou, B., Wilson, J., Zhan, F., & Zeng, Y. (2009). Air pollution exposure assessment methods utilized in epidemiological studies. Journal Of Environmental Monitoring, 11(3), 475. doi: 10.1039/b813889c
However, the correlation coefficient (r) was explored linear relationship between two variables of pollution and meteorological variable among 3 provinces. Values more than 0.70 indicate a very strong positive relationship, 0.40 and 0.69 indicate a strong positive relationship, 0.30 to 0.39 indicate a moderate positive relationship, 0.20 to 0.29 indicate a strong positive relationship. The correlation coefficient was very strong positive relationship for relative humidity (0.71), temperature (0.76-0.77), strong-very strong positive relationship for pm10 (0.58-0.79), NO2 (0.43-0.71) and O3 (0.46-0.82) except for SO2 was weak negative correlation (-0.09-2.5). Therefore, the data from one station could be sufficient for this analyse
Therefore, we added the sentence in the limitation as follows;
“the available ambient air pollution and meteorological variable from only one fixed-site air monitoring station in Chachoengsao was assigned as an exposure level to the population. These exposure misclassifications can cause a bias in the measurement of affect toward the null value. It produces an underestimate of the effect; however, it is unavoidable [56,57]”
as shown on the revised manuscript on line 590
Point 5:
Line 108, how many/ percentage of days or data points are labeled as “missing” based on these criteria? Can the authors also elaborate on how the EM algorithm was used to impute the missing values and why would this be optimal? What statistic/distribution model was used as an assumption of pollution values?
Response 5:
The percentage of the missing days was shown in this table
|
Provinces (2556 days) |
PM2.5 |
PM10 |
NO2 |
SO2 |
O3 |
CO |
|
Chachoengsao |
2,305 (90.18%) |
255 (9.98%) |
539 (21.09%) |
336 (13.15%) |
265 (10.37%) |
2,031 (79.46) |
|
Chonburi |
911 (35.64%) |
3 (0.12%) |
1 (0.04%) |
179 (7.00%) |
0 (0.00%) |
93 (3.64%) |
|
Rayong |
122 (4.77%) |
0 (0.00%) |
3 (0.12%) |
1 (0.04%) |
0 (0.00%) |
0 (0.00%) |
A higher proportion of missing observations may result in a loss of precision. In time series analysis, this problem can be exacerbated because excluding missing observations may corrupt temporal structures such as autocorrelation, trends, and seasonality. Therefore, the expectation-maximization (EM) algorithm method was applied to impute of missing data in multivariate normal time series, which uses the method under the assumption of multivariate normal distribution. For example, the PM2.5 imputation equation was
PM2.5 = ns(NO2, 3)+ns(SO2, 3)+ns(PM10, 3)+ns(CO, 3)+ns(RH, 3)
This algorithm is specially tailored for climate data with missing measurements from several monitors along a given region. The EM algorithm methods have been implemented as a package called mtsdi (multivariate time series data imputation) for the statistical software R.
According to the study of Junger, W. and Ponce De Leon, A. (2015) showed the estimates of the simulation, using 40% missing data under missing completely at random (MCAR) using the EM with a temporal component and multiple covariances filtered by splines (EM-MV spline) were a low value of root mean square error (RMSE) and mean absolute error (MAE) and high value of Pearson's correlation coefficient (r). Therefore, if air pollution and meteorological data were missing for less than 25% of the study period, the RMSE and MAE would be lower, and the r would be higher than 40% missing. The missing data were replaced with values estimated using the EM algorithm. If air pollution and meteorological data were missing for more than 25% of the study period, the missing data were excluded from the study.
Junger, W., & Ponce De Leon, A. (2015). Imputation of missing data in time series for air pollutants. Atmospheric Environment, 102, 96–104. https://doi.org/10.1016/j.atmosenv.2014.11.049
Therefore, we added the sentence to explain on how the EM algorithm was used to impute the missing values in the methods as follows;
“..., which uses the method under the assumption of multivariate normal distribution. This algorithm is specially tailored for climate data with missing measurements from several monitors along a given region ...”
as shown on the revised manuscript on line 185
Point 6:
Line 120, how many hospitals (for each region) were involved in this study?
Response 6:
There were 1,226 hospitals including government hospitals, private hospitals, primary care health units, and private clinics located in the ECC, 239 in Chachoengsao, 430 in Chonburi, and 557 in Rayong. The data of OPD visits patient who has the Universal Coverage Scheme (UCS) were reported to the National Health Security Office (NHSO). The data for OPD visits for various causes (including cardiovascular diseases, respiratory diseases, and skin and subcutaneous tissue diseases) in the EEC during the period air pollution and meteorological data were acquired by NHSO
Mortality data derived from information reported on death certificates during the same period were supplied by the Strategy and Planning Division of the Thailand Ministry of Public Health.
Therefore, we added the information and restatement to explain on how we got the outpatient department visits and mortality data as follows;
“There were 1,226 hospitals including government hospitals, private hospitals, primary care health units, and private clinics located in the ECC, 239 in Chachoengsao, 430 in Chonburi, and 557 in Rayong. Data of patient who has the Universal Coverage Scheme (UCS) were reported to the National Health Security Office (NHSO). The data for OPD visits for various causes (including cardiovascular diseases, respiratory diseases, and skin and subcutaneous tissue diseases) in the EEC during the period air pollution and meteorological data were acquired by NHSO. The visit date, the date of birth, age, and sex of the patient, and the primary diagnosis code from the International Classification of Diseases 10th Revision were acquired. Mortality data derived from information reported on death certificates during the same period were supplied by the Strategy and Planning Division of the Thailand Ministry of Public Health. The date of birth, age, and sex of each deceased person and the International Classification of Diseases 10th Revision cause-of-death code were used.”
as shown on the revised manuscript on line 190
Point 7:
Line 122, stated that the collected data contains age and gender information. However, in the subsequent analysis, no analysis was shown on related categories ( eg, pediatric visits vs adult visits, male vs female). Is there any specific reason such analysis was not performed?
Response 7:
Because we want to examine the overall population not specifically to the different practically sub-specific population. The result can represent all populations groups in this area but not a specific group. However, the main objective of this study aimed to examine the association between exposure to daily changes in ambient air pollution and OPD visits, and mortality for a spectrum of causes in overall population in the EEC but not specifically in age or gender. Therefore, this study could not show the association in the risky group and the susceptible group. Moreover, our aspect was to understand and get evidence on adverse health effects related to air pollution situation in area-based development, where is a location for manufacturing, as a based-line data to strengthen the policy and preventative intervention to further minimizing the adverse health effects of air pollution and protect population health. The estimation of the association between exposure to air pollution and population health in specific age group and sex will suggest studying for the future study.
We added the sentence in the limitation as follows;
“... Fifth, the results represented all populations in the EEC but not a specific subgroup in age or gender. The risky group or the susceptible group should be analysed in the future...”
as shown on the revised manuscript on line 598
Point 8:
Line 148, can the authors elaborate on why these variables (eg temperate, humidity) are sufficient to control the effects of air pollution on hospital visits? How about local populations etc.
Response 8:
We explored how a model's predictive accuracy affects by using the forward stepwise selection method, which starts with a model containing no variables and then adds the most significant variables one after the other. We used the Akaike Information Criterion (AIC) to evaluate how well a model fits with the data. The smaller the AIC value, the better the model fit. The temperate and humidity made the lower AIC value for this model. Then, we used those two variables (temperate and humidity) to control the effects of air pollution on OPD visits and mortality. Although, temperate and humidity variables affected a model's predictive accuracy, but the local population wasn’t classified as a confounder because this study was examined the short-term effect using a time-stratified case-crossover design, a matched case–control within the same day of the week in the same calendar month of the same year. There wasn't a difference in population in each stratum. Therefore, the population wasn’t a confounder for this model
Point 9:
Line 197, “the” fewest in Chachoengsao….
Response 9:
Thank you. We do accept the suggestion and already edited them as shown on line 329
Point 10:
Line 202, in the order “of” circulatory diseases…
Response 10:
Thank you. We do accept the suggestion and already edited them as shown on line 334
Point 11:
Line 245, table 4 title unit increase in pollutant concentrations… (no “the”), same for table 5 title.
Response 11:
Thank you. We do accept the suggestion and already edited them as shown on line 379 and 383
Point 12:
Line 246, table 4. In this table, the authors presented the short-term association of air pollution to various hospital visits, with the relative risk values. Since there are some similar literature that also investigated this association in Thailand, how does the finding compare to other regions in Thailand or southeast Asia? Among those associations, is there anything specific in the Eastern Economic Corridor region? For example, Is the relative risk higher in the Eastern Economic Corridor region compared to other places?
Response 12:
We added the sentence in the discussion as follows;
“at lag0, the RR of respiratory outpatients increased by 0.37% with a 10 μg/m3 increase in PM2.5 in Shanghai, China [44]. In addition, the studies performed in, Thailand showed the association between a 10 μg/m3 increase in PM10 concentration and increase cardiovascular mortality (1.90% 95% CI 1.00, 2.80) [10]. In Bangkok Metropolitan Region, Odds ratio of respiratory OPD among children per 10 µg/m3 increase in PM10 and PM2.5 was 1.03 (95% CI 1.02, 1.03) and 1.03 (95% CI 1.01, 1.06), respectively [11]. The study of the impact of PM on daily mortality in Bangkok indicated a statistically significant association between PM10 and cardiovascular and respiratory mortality, a 10 µg/m3 change in daily PM10 is associated with a 1–2% and a 3–6% increase, respectively [39]. The estimated effect seems higher than this study because the estimation was explored in 10 µg/m3.”
as shown on the revised manuscript on line 509
Point 13:
Line249, table 5. Since the association evaluated was focused on the short-term exposure, using the case-crossover method, I’m surprised that the relative risk is still high and significant to mortality rates. This means exposure to increased air pollution as short as one day, could lead to direct death. Has the same short-term association of mortality been found in other literature? If not or rarely, does this means the current model used in the paper is possibly exaggerating the effect in some way?
Response 13:
We added the sentence in the discussion as follows;
“Higher relative risk of short-term exposure to air pollution on mortality may be explained by the susceptible population reasons. Older adults and children or people with heart (or lung) disease are subject to much stronger risk from particles than other people [45]. Several epidemiologic studies have shown the evidence of short-term exposure to air pollution on mortality. The estimated effect of short-term exposure to PM10 and PM2.5 from meta-analysis studies of 17 Chinese cities shown a10 μg/m3 increase in PM2.5 concentration associated with an increase of mortality due to circulatory diseases, respiratory diseases were 0.63% (95% CI 0.35, 0.91%), and 0.75% (95% CI 0.39, 1.11%), respectively [46]. the study in the Netherlands, 10 μg/m3 increases in PM2.5 were related with an increase of 1.60% (95% CI 0.40, 2.90%) in respiratory mortality, 1.10%, (95% CI 0.20, 1.90%) in Cardiovascular at lag0 [47]. Short-term PM2.5 exposure were significantly associated with increase all-cause mortality among residents of New England ≥ 65 years of age, 2.14% (95% CI 1.38, 2.89%) increases were estimated for each 10 μg/m3 increase in short-term exposure (2 day). even for exposure particle exposure levels below the standards established by the United States, the World Health Organization (WHO) (25 μg/m3 of daily average PM2.5) [48]. The study in Thailand shown the high cardiovascular mortality percent change of a 10 μg/m3 increase in PM10 [1.90% (95% CI 1.00, 2.80)] at lag0 [10]. The mechanisms of PM on triggering respiratory diseases show that the oxidative stress and inflammation process driving adverse effects related to respiratory diseases. Particularly, PM could trigger inflammatory response, potentially leading to cause airway obstruction, affect gas-exchange process and exacerbate pre-existing condition of respiratory disease [49,50].”
as shown on the revised manuscript on line 520
Point 14:
Line 250, following the same question, if the authors analyze the data with another model, eg. Looking at lag days (polynomial distributed lag models etc), does the association (hospital visits and mortality) still hold?
Response 14:
We analyzed distributed lag non-linear models up to lag7. The best lag was observed at the current lag (lag0). The result related to previous studies in which are the best lag was on the current lag.
Zhang, Y., Ni, H., Bai, L., Cheng, Q., Zhang, H., Wang, S., Xie, M., Zhao, D., & Su, H. (2019). The short-term association between air pollution and childhood asthma hospital admissions in urban areas of Hefei City in China: A time-series study. Environmental Research, 169, 510–516. https://doi.org/10.1016/j.envres.2018.11.043
Zhu, X., Qiu, H., Wang, L., Duan, Z., Yu, H., Deng, R., Zhang, Y., & Zhou, L. (2019). Risks of hospital admissions from a spectrum of causes associated with particulate matter pollution. Science of The Total Environment, 656, 90–100. https://doi.org/10.1016/j.scitotenv.2018.11.240
Point 15:
Line 289, table 9 title, Thailand “exceeded” the national ambient air quality standard…
Response 15:
We have now changed ‘exceeding’ to ‘exceeded’ throughout the manuscript.

Reviewer 2 Report
Please see the attached file.

Author Response
Response to Reviewer 2 Comments
Comments and Suggestions for Authors
This manuscript investigated the association between air pollutants exposure and human health in EEC. My recommendation is to accept after minor revision. My question or comment are shown below
Point 1:
Although this is a first study to investigate the association between air pollutants exposure and health effects in EEC, many studies have demonstrated that air pollutant exposure is an important risk factor to human health, including in heavy industrial areas. I don't see any new scientific information through this study. The authors should clearly describe what new scientific information their study provide.
Response 1:
Thank you for the insightful comments. We agreed with the reviewer and described what new scientific information in this study.
The new finding in this study has shown the benefits of reducing the ambient air pollution concentration of the National Ambient Air Quality Standards (NAAQSs) in Thailand could reduce the health effect on the Thais population. The results from the study will be evidence to strengthen the policy for reducing the ambient air pollution concentration and preventative intervention to further minimize the adverse health effects of air pollution and protect population health in particular in industrial areas
We added the sentences to clarify the new scientific information in the abstract as follows; “Preventing PM10 concentrations from being higher than 60 µg/m3 could decrease more than 33,265 and 29,813 OPD visits for circulatory and respiratory diseases, respectively. Our study suggests that those pollution increase risks of OPD visits and mortality in various causes in the Thai EEC. The reducing the ambient air pollution concentration of NAAQSs in Thailand could reduce the health effect on the Thais population.”
as shown on the revised manuscript on line 32
Point 2:
In lines 98 to 99, how to confirm the representative of air pollution and meteorological data to population exposure in EEC. For example, in Chachoengsao, only one monitoring station.
Response 2:
There was one of the limitations, the available ambient air pollution and meteorological variable from only one fixed-site air monitoring station in Chachoengsao was assigned as an exposure level to the population. These would be exposure misclassification causes measures of effect to be biased toward the null value. misclassification produces an underestimate of the effect, whereas misclassification can result in bias either toward or away from the null value, is unavoidable.
National Academies of Sciences, Engineering, and Medicine. 1997. Environmental Epidemiology, Volume 2: Use of the Gray Literature and Other Data in Environmental Epidemiology. Washington, DC: The National Academies Press. https://doi.org/10.17226/5804.
Zou, B., Wilson, J., Zhan, F., & Zeng, Y. (2009). Air pollution exposure assessment methods utilized in epidemiological studies. Journal Of Environmental Monitoring, 11(3), 475. doi: 10.1039/b813889c
However, the correlation coefficient (r) was explored linear relationship between two variables of pollution and meteorological variable among 3 provinces. Values more than 0.70 indicate a very strong positive relationship, 0.40 and 0.69 indicate a strong positive relationship, 0.30 to 0.39 indicate a moderate positive relationship, 0.20 to 0.29 indicate a strong positive relationship. The correlation coefficient was very strong positive relationship for relative humidity (0.71), temperature (0.76-0.77), strong-very strong positive relationship for pm10 (0.58-0.79), NO2 (0.43-0.71) and O3 (0.46-0.82) except for SO2 was weak negative correlation (-0.09-2.5). Therefore, this data from one station could sufficient for analyse
Therefore, we added the sentence in the limitation as follows;
“the available ambient air pollution and meteorological variable from only one fixed-site air monitoring station in Chachoengsao was assigned as an exposure level to the popu-lation. These exposure misclassifications can cause a bias in the measurement of effect toward the null value. It produces an underestimate of the effect; however, it is una-voidable [56,57].”
as shown on the revised manuscript on line 602
Point 3:
This study did not analyze or discuss the effect of Barometric pressure on their outcome. Thus, I suggest that the author should delete the Barometric pressure data in the main text.
Response 3:
Thank you for your suggestion. However, the barometric pressure was not included for analysis in this study. All barometric pressure mentioned in this study has been removed.

Reviewer 3 Report
The paper entitled “Outpatient Department Visits and Mortality with Various Causes Attributable to Ambient Air Pollution in the Eastern Economic Corridor of Thailand” deals OPD attributed PM2.5 and PM10, CO, Ozone, SO2 and NO2. Such studies are a few from this region and therefore these results are important. However, there are numerous issues that should be taken care of. Overall manuscript needs to be revised thoroughly, I suggest major revision. My comments are as follows:
1. Abstract is very general not provide findings of present work
2. Please highlight objectives in introduction and also what is new in this work.
3. Please provide the real time concentration of study area (if it available in literature).
4. Please provide detail description of location (latitude and longitude coordinates, distance from sea level, area population density etc. I suggest total population is fine (information regarding migrants is seem irrelevant here.
5. Have authors used online source for data? If yes provide information
6. How they calculate coefficient β1 and β2 of each OPD
7. Line 153 “pair on day t, ns() is the natural cubic spline…..” correct it
8. Why author did not access widespread and wind direction? What about transboundary movement of pollutants?
9. Authors did not discuss CO, Ozone and SO2 based OPD in discussion section (if findings are insignificant, at least provide a single statement.
10. Line 336-338: authors stated “We found higher daily mean NO2 concentrations, PM10 concentrations, CO concentrations, and temperatures during the study period in Chonburi than the other provinces” they should discuss the reasons of this difference.
11. Similarly, they only provide information regarding contrasting OPD visits with PM10 circulatory diseases, respiratory skin and subcutaneous tissue diseases and PM2.5 death from circulatory diseases, respiratory diseases skin and subcutaneous tissue diseases, they should provide reasonable justification of these trend (based on epidemiology studies).
12. In limitation they should mentioned if they have accessed change in population during these years? How they eliminate impact of gender and age?
13. The topic and introduction suggest that the study will also debate Economic Corridor factor however author did discuss it, I suggest make it little more relevant or remove this expression.
Author Response
Response to Reviewer 3 Comments
Comments and Suggestions for Authors
The paper entitled “Outpatient Department Visits and Mortality with Various Causes Attributable to Ambient Air Pollution in the Eastern Economic Corridor of Thailand” deals OPD attributed PM2.5 and PM10, CO, Ozone, SO2 and NO2. Such studies are a few from this region and therefore these results are important. However, there are numerous issues that should be taken care of. Overall manuscript needs to be revised thoroughly; I suggest major revision. My comments are as follows:
Point 1:
Abstract is very general not provide findings of present work
Response 1:
Thank you for the insightful comments. We agreed with the reviewer and described what new scientific information in this study. The new finding in this study has shown the benefits of reducing the ambient air pollution concentration of the National Ambient Air Quality Standards (NAAQSs) in Thailand could reduce the health effect on the Thais population. The results from the study will be epidemiological evidence to strengthen the policy for reducing the ambient air pollution concentration and preventative intervention to further minimize the adverse health effects of air pollution and protect population health in particular in industrial areas
Then we revised abstract as show below
“The Eastern Economic Corridor in Thailand is undergoing development, but industrial activities are causing serious air pollution. This study aimed to examine the effects of particulate matter (PM10), fine particulate matter (PM2.5), SO2, NO2, O3, and CO on outpatient department (OPD) visits and mortality with various causes in the Eastern Economic Corridor, Thailand during 2013-2019, using a case-crossover design and conditional Poisson model. The corresponding burden of disease due to air pollution exposure was calculated. A 1 µg/m3 increase in the PM10 was associated with significant increases in OPD visits for circulatory diseases (0.22, 95% CIs 0.01, 0.34), respiratory diseases (0.21, 95% CIs 0.13, 0.28), and skin and subcutaneous tissue diseases (0.18, 95% CIs 0.10, 0.26). By contrast, a 1 µg/m3 increase in the PM10 was associated with significant increases in mortality only skin and subcutaneous tissue diseases (0.79, 95% CIs 0.04, 1.56). a 1 µg/m3 increase in PM2.5 was associated with significant increases in mortality from circulatory diseases (0.75, 95% CIs 0.20, 1.34), respiratory diseases (0.82, 95% CIs 0.02, 1.63), and skin and subcutaneous tissue diseases (2.91, 95% CIs 0.99, 4.86). The highest OPD burden was for circulatory diseases. Respiratory effects were attributed to PM10 exceeding the national ambient air quality standards (NAAQS) of Thailand (120 μg/m3). The highest morbidity burden was for skin and subcutaneous tissue diseases attributed to PM2.5 concentrations exceeding the NAAQs (50 μg/m3). PM pollution in the EEC could strongly contribute to OPD visits and morbidity from various diseases. Preventing PM10 concentrations from being higher than 60 µg/m3 could decrease more than 33,265 and 29,813 OPD visits for circulatory and respiratory diseases, respectively. Our study suggests that those pollution increase risks of OPD visits and mortality in various causes in the Thai EEC. The reducing the ambient air pollution concentration of NAAQSs in Thailand could reduce the health effect on the Thais population.”
as shown on the revised manuscript on line 16
Point 2:
Please highlight objectives in introduction and also what is new in this work.
Response 2:
The main objective of this study aimed to examine the association between exposure to daily changes in ambient air pollution and OPD visits, and mortality for a spectrum of causes in overall population in the EEC.
The new finding in this study has shown the benefits of reducing the ambient air pollution concentration of the National Ambient Air Quality Standards (NAAQSs) in Thailand could reduce the health effect on the Thais population. The results from the study will be evidence to strengthen the policy for reducing the ambient air pollution concentration and preventative intervention to further minimize the adverse health effects of air pollution and protect population health in particular in industrial areas.
We added a statement as the following sentense;
“The main objective of this study aimed to examine the association between exposure to daily changes in ambient air pollution and OPD visits, and mortality for a spectrum of causes in overall population in the EEC.”
as shown on the revised manuscript on line 127
Point 3:
Please provide the real time concentration of study area (if it available in literature).
Response 3:
Thank you, the real-time air quality data were measured at fixed ambient air monitoring stations operated by the Thailand Pollution Control Department (Fig.1) and are available only on request, not publicly available.
Point 4:
Please provide detail description of location (latitude and longitude coordinates, distance from sea level, area population density etc. I suggest total population is fine (information regarding migrants is seem irrelevant here.
Response 4:
Thank you for the insightful comments. We agreed with the reviewer and provided the detail of this location as the following sentence;
“Thailand Eastern Economic Corridor (EEC) is an area-based development initiative, initially focused on the 3 Eastern provinces of Thailand because of their strategic locations near deep seaports and natural resources in the Gulf of Thailand, including Rayong, Chonburi, and Chachoengsao provinces and spans approximately 13,266 sq.km., as shown in Figure 1. The EEC plays an important role in Thailand's fast-growing economy aim at transforming the country into a hub for technological manufacturing and services with strong connectivity to its association of Southeast Asian Nations (ASEAN) neighbors by land, rail, sea and air transportation. …”
as shown on the revised manuscript on line 132
The slow gradual change in population was simply assumed to be the same as every single year.
And added a statement as the following sentence;
“Most of the population lives in urban areas where have a monitoring station. Therefore, the population density was high in urban areas.”
as shown on the revised manuscript on line 155
Point 5:
Have authors used online source for data? If yes provide information
Response 5:
We haven’t used any online data source for the analysis. All relevant data were obtained from each responsible agency. We submitted the request form to each responsible agency consisting of OPD visits data were obtained from the National Health Security Office (NHSO), Mortality data were retrieved from the strategy and planning division, ministry of public health, and air pollution and meteorological data were obtained from Thailand’s Pollution Control Department (PCD) are available only on request, not publicly available.
Point 6:
How they calculate coefficient β1 and β2 of each OPD
Response 6:
β1 is vector of coefficient for each air pollution associated with OPD/Mortality. We wrote the wrong equation. We do apologize for incorrect equation. In this case, we put indicator variable of province.
We revised as show in the equation 1 on the revised manuscript on line 229
“Log(E(Yt)) = α + β1Xt + ns(tempt, 3df) + ns(RHt, 3df) + provincet + stratumt,”
Point 7:
Line 153 “pair on day t, ns() is the natural cubic spline…..” correct it
Response 7:
We revised the sentences as show...;
“... ns() is the natural cubic spline with a defined number of degrees of freedom (df), and tempt and RHt are the temperature and relative humidity on day t, provincet is indicator variable on day t, stratum is a matched case–control within the same day of the week in the same calendar month of the same year on day t, respectively.”
as shown on the revised manuscript on line 287
Point 8:
Why author did not access widespread and wind direction? What about transboundary movement of pollutants?
Response 8:
We did not adjust for widespread and wind direction because of data unavailability. We explored how a model's predictive accuracy affects by using the forward stepwise selection method, which starts with a model containing no variables and then adds the most significant variables one after the other. We used the Akaike Information Criterion (AIC) to evaluate how well a model fits with the data. The smaller the AIC value, the better the model fit. The temperate and humidity made the lower AIC value for this model. Then, we used those two variables to control the effects of air pollution on OPD visits and mortality.
Point 9:
Authors did not discuss CO, Ozone and SO2 based OPD in discussion section (if findings are insignificant, at least provide a single statement.
Response 9:
Although, A 1 ppb increase in the NO2 concentration was associated with significant increases in OPD visits for circulatory diseases (1.38, 95% CIs 1.04 and 1.72), respiratory diseases (0.93, 95% CIs 0.73 and 1.14), and skin and subcutaneous tissue diseases (1.03, 95% CIs 0.79 and 1.27) and a 1 ppb increase in the SO2 concentration was associated with significant increases in OPD visits for respiratory diseases (1.23, 95% CIs 0.73 and 1.74) and skin and subcutaneous tissue diseases (0.71, 95% CIs 0.15 and 1.27), attributable fractions and attributable numbers of OPD visits for circulatory diseases, respiratory diseases, and skin and subcutaneous tissue diseases attributable to NO2, SO2 were small amount. The associations between these pollutions and health outcomes in the EEC should be study in the further with difference approaches.
We added a statement as the following sentense;
“Although, A 1 ppb increase in the NO2 concentration was associated with significant increases in OPD visits for circulatory diseases, respiratory diseases, and skin and subcutaneous tissue diseases and a 1 ppb increase in the SO2 concentration was associated with significant increases in OPD visits for respiratory diseases and skin and subcutaneous tissue diseases, attributable fractions and attributable numbers of OPD visits for circulatory diseases, respiratory diseases, and skin and subcutaneous tissue diseases attributable to NO2, SO2 were small amount in this study. The associations between these pollutions and health outcomes in the EEC should be study in the further with difference approaches”
as shown on the revised manuscript on line 579
Point 10:
Line 336-338: authors stated “We found higher daily mean NO2 concentrations, PM10 concentrations, CO concentrations, and temperatures during the study period in Chonburi than the other provinces” they should discuss the reasons of this difference.
Response 10:
Chonburi is a rapidly growing economy and rising in population and urbanization compared with other provinces. Considering a stationary source as a factory and industrial estate, there were 5,418 factories registered and 11 industrial estates in the province, the highest proportion of the factory and industrial estate were 50.42% and 52.38%, respectively [18, 19]. Moreover, mobile sources as a vehicle, there were 1,628,404 cars registered (57.76%). These point sources could lead higher air pollution concentration than other provinces.
Department of Land Transport. statistics. Available online: https://web.dlt.go.th/statistics/ (in Thai) (accessed on 14 June 2022).
We added a statement as the following sentense;
“Chonburi is a rapidly growing economy and rising in population and urbanization compared with other provinces. Considering to a stationary source as a factory and industrial estate, there were 5,418 factories registered and 11 industrial estates in the province, the highest proportion of the factory and industrial estate were 50.42% and 52.38%, respectively [18, 19]. Moreover, mobile sources as a vehicle, there were 1,628,404 cars registered (57.76%) [42] These point sources could lead higher air pollution concentration than other provinces.”
as shown on the revised manuscript on line 486
Point 11:
Similarly, they only provide information regarding contrasting OPD visits with PM10 circulatory diseases, respiratory skin and subcutaneous tissue diseases and PM2.5 death from circulatory diseases, respiratory diseases skin and subcutaneous tissue diseases, they should provide reasonable justification of these trend (based on epidemiology studies).
Response 11:
Unfortunately, data on PM2.5 were available only in Rayong province. Data on PM10 among 3 provinces were considered as a result of the larger sample size than PM2.5, which available only in Rayong province then the sample size would less than PM10. The error of PM2.5 would be greater than PM10. Therefore, The association between PM2.5 and OPD for a spectrum of causes was not observed. the association between PM10 and mortality for a spectrum of causes was not observed. The point estimate was positive association, but 95% CI were not significant because the wider in error then wider in 95% CI
Point 12:
In limitation they should mentioned if they have accessed change in population during these years? How they eliminate impact of gender and age?
Response 12:
This study was examined the short-term effect using a time-stratified case-crossover design, all the study subjects are cases, each subject serves as his or her own control, a matched case–control within the same day of the week in the same calendar month of the same year. There wasn't a difference in population in each stratum. Therefore, the population wasn’t a confounder for this model.
Point 13:
The topic and introduction suggest that the study will also debate Economic Corridor factor however author did discuss it, I suggest make it little more relevant or remove this expression.
Response 13:
We decided to remove it as reviewer suggestion.

Round 2
Reviewer 3 Report
I am pleased that the authors have improved their manuscript in satisfactory ways. However, some minor issues should be addressed as follow.
Please read the manuscript carefully and correct typo mistakes such as line 25 (“a” 1 µg, correct it). Similarly, pay attention to subscript and superscripts, for instance, lines 31 “μg/m3” and 32 “PM10” and so on.
In my previous comment, I asked about the values of “β1”. Authors should provide the values of the vector of coefficient that they used in the given equation and the sources from which they obtained or how they calculated the values to use in the equation to calculate each air pollution associated with OPD/Mortality.
In response to my Point 11: the authors provide reasonable justification; however, it should be mentioned in the limitation part of this study starting from line 495.
Author Response
Response to Reviewer 3 Comments
Comments and Suggestions for Authors
I am pleased that the authors have improved their manuscript in satisfactory ways. However, some minor issues should be addressed as follow.
Point 1:
Please read the manuscript carefully and correct typo mistakes such as line 25 (“a” 1 µg, correct it). Similarly, pay attention to subscript and superscripts, for instance, lines 31 “μg/m3” and 32 “PM10” and so on.
Response 1:
Thank you for the insightful comments. We agreed with the reviewer and edited as suggestion;
“A 1 µg/m3 increase….” as shown in line 25
“μg/m3” as shown in line 30-31
“PM10” as shown in line 32
Point 2:
In my previous comment, I asked about the values of “β1”. Authors should provide the values of the vector of coefficient that they used in the given equation and the sources from which they obtained or how they calculated the values to use in the equation to calculate each air pollution associated with OPD/Mortality.
Response 2:
Thank you for your suggestion and we have provided the values of the vector of coefficient in Table S1 and S2 for each air pollution associated with OPD/Mortality.
For example, the associations between 1 µg/m3 increases in PM10 concentrations and OPD visits for circulatory diseases or β1 per 1 µg/m3 increase in PM10 was calculated from this equation;
Log(E(Yt)) = α + β1PM10t + ns(tempt, 3df) + ns(RHt, 3df) + provincet + stratumt,
and then β1 was 0.002198 as shown in Table S1
The associations between 1 µg/m3 increases in PM2.5 concentrations and mortality for respiratory diseases or β1 per 1 µg/m3 increase in PM2.5 was calculated from the same equation as follows;
Log(E(Yt)) = α + β1PM2.5t + ns(tempt, 3df) + ns(RHt, 3df) + provincet + stratumt,
and then β1 was 0.008167 as shown in Table S2
Therefore, we added the sentence to provide the values of the vector of coefficient as follows;
“β1 is the regression coefficient per unit increase in the air pollutant concentration and the values of the vector of coefficient were presented in Table S1and S2”
as shown on the revised manuscript in line 234
We also calculated each β1 by the maximum likelihood method. Therefore, we added the sentence as follows;
“β1 was calculated by the maximum likelihood method”
as shown on the revised manuscript in line 235
Point 3:
In response to my Point 11: the authors provide reasonable justification; however, it should be mentioned in the limitation part of this study starting from line 495.
Response 3:
Thank you for the insightful comments. We agreed with the reviewer and mentioned it in the limitation part of this study as follows;
“The study had some limitations. First, PM2.5 data for Chachoengsao and Chonburi were insufficient available and inappropriate for the study period. Therefore, those data were excluded from the study. Unfortunately, data on PM2.5 were available only in Rayong. Consequently, the association between PM2.5 and OPD for a spectrum of causes was not statistically significant. More PM2.5 data will be required for further study.”
as shown on the revised manuscript in line 755
